# Characterizing the Dynamic Expression of C1q/TNF-α-Related Protein 6 (CTRP6) during Pregnancy in Humans and Mice with Gestational Diabetes Mellitus

**DOI:** 10.3390/biomedicines12051128

**Published:** 2024-05-19

**Authors:** Jianan Jiang, Shuangyu Wei, Miao Chen, Yutian Tan, Zhao Yang, Guiying Yang, Weijie Feng, Zhen Han, Xiaojing Wei, Xiao Luo

**Affiliations:** 1Department of Physiology and Pathophysiology, School of Basic Medical Sciences, Xi’an Jiaotong University Health Science Center, Xi’an 710061, China; jiangjianan0409@stu.xjtu.edu.cn (J.J.); tanyutian@stu.xjtu.edu.cn (Y.T.); fwj001229@126.com (W.F.); 2Institute of Neuroscience, Translational Medicine Institute, Xi’an Jiaotong University Health Science Center, Xi’an 710061, China; 3Key Laboratory of Environment and Genes Related to Diseases, Ministry of Education of China, Xi’an Jiaotong University, Xi’an 710061, China; 4Clinical Laboratory, The First Affiliated Hospital, Xi’an Jiaotong University, Xi’an 710061, China; watashinoshinichi@126.com; 5Department of Obstetrics and Gynecology, The First Affiliated Hospital, Xi’an Jiaotong University, Xi’an 710061, China; 004844@xjtu.edu.cn (M.C.); yangzhao123@stu.xjtu.edu.cn (Z.Y.); yanggy0912@stu.xjtu.edu.cn (G.Y.); hanamy02@163.com (Z.H.)

**Keywords:** gestational diabetes mellitus, CTRP6, plasma, adipose tissue, placenta

## Abstract

Aim: C1q/TNF-related protein 6 (CTRP6) is a novel adipokine involved in insulin resistance. Thus, we aim to investigate the expression profile of CTRP6 in the plasma, adipose tissue and placenta of GDM patients and mice. Methods: Chinese Han pregnant women (GDM *n* = 9, control *n* = 10) with a scheduled caesarean section delivery were recruited. A number of high-fat diet (HFD) induced-pregnancy C57BL/6 mice were chosen as an animal model of GDM. Circulating levels of CTRP6 and adiponectin were examined by ELISA. CTRP6 expression in adipose tissue and placenta were detected by real-time qPCR and WB. Result: Plasma CTRP6 levels were decreased during the first and second trimesters in mice, as well as the second and third trimesters in patients, while they were increased at delivery in GDM patients and mice. Plasma CTRP6 levels were significantly correlated with WBC, systolic pressure, diastolic pressure and fasting blood glucose. Moreover, CTRP6 mRNA expression in the subcutaneous (sWAT) and omental white adipose tissue (oWAT), as well as in the placenta, was significantly higher in GDM human patients at cesarean delivery. Furthermore, the mRNA expression of Ctrp6 was increased in the sWAT and visceral WAT (vWAT), whilst decreased in the interscapular brown adipose tissue (iBAT), of GDM mice at cesarean delivery. Conclusion: Dynamically expressed CTRP6 may be served as a candidate target for treatment of GDM.

## 1. Introduction

Gestational Diabetes Mellitus (GDM) refers to abnormal glucose tolerance of different degrees with onset or first recognition during pregnancy [1]. The epidemiology prevalence of GDM has significantly increased in recent years due to risk factors including the increase in rates of overweight or obesity in women of reproductive age and rising maternal age [2]. GDM is a high-risk pregnancy that is associated with obstetric and neonatal complications relating to macrosomia and postpartum hemorrhage and is increasingly recognized as a risk factor for future maternal and offspring metabolic disease [3,4]. GDM usually develops from the second trimester of pregnancy with insulin resistance and/or β-cell disruption as typical pathological feature [2]. Although the precise mechanism of GDM insulin resistance is unclear, a large number of studies have shown that adipose tissue, as one of the insulin target organs, plays an important role in regulation of insulin sensitivity [5,6,7]. 

Adipose tissue does not only store triglycerides; it is also an important endocrine organ of the body. Through secreting a large number of adipokines with different functions, it participates in regulating maternal insulin sensitivity during pregnancy, placental nutrient transport, and maintaining the balance of maternal and fetal energy metabolism [8]. To date, it is well known that several adipokines, including leptin, adiponectin, resistin, visfatin and tumor necrosis factor α (TNFα) are dysregulated in GDM and contribute to maternal metabolic complications [9,10]. As the first reported adipokine, leptin abnormally increased in circulating levels in GDM women compared to the normal glucose tolerance (NGT) population [11]. It was suggested that leptin acts as a circulating signal to control fetal homoeostasis, and higher leptin levels may increase the risk of obesity in offspring exposed to GDM [11]. Meanwhile, adiponectin is notable for its action in mediating insulin sensitivity. To date, many studies have shown that adiponectin levels decrease in GDM women compared to NGT controls, indicating that its level could be a predictive factor for the development of GDM [9]. Indeed, the leptin/adiponectin ratio may be used as a predictor of GDM risk at the first trimester of pregnancy [12]. 

C1q/tumor necrosis factor-related protein 6 (CTRP6) is a novel adipokine which contains 240 amino acids and is secreted as oligomer in serum [13]. It is widely expressed in adipose tissue and placenta as well as in the heart [13]. As an adiponectin paralog protein, CTRP6 is involved in several physiological or pathological processes, such as rheumatoid arthritis, diabetes and cardiovascular disease [14,15]. CTRP6 inhibits the proliferation of rat mesangial cells and the expression of extracellular matrix (ECM) [16]. In addition, a previous study in mice revealed that in vivo supplementation of CTRP6, as a complement regulator, can improve intrauterine fetal growth restriction [17]. Another clinical study showed a significant increase in serum CTRP6 in patients with polycystic ovary syndrome [14]. A recent study in recurrent spontaneous abortion has shown that CTRP6 regulates the polarization of M1 macrophages through the PPAR-γ/NF-κB pathway and glycolytic reprogramming [18]. Furthermore, the increase in serum CTRP6 levels was observed in Type 2 diabetes mellitus (T2DM) and impaired glucose tolerant individuals compared with healthy controls [19]. Over-expression of CTRP6 was found to reduce insulin-stimulated Akt phosphorylation and glucose uptake in mice and adipocytes [20]. CTRP6 knockdown inhibited diet-induced obesity and improved insulin sensitivity in mice [21]. However, compared with the studies on obesity and T2DM, there are relatively few studies on the role of CTRP6 in GDM, although obesity has been generally considered to be inseparable from GDM. Recently, CTRP6 was found to regulate the viability, migration and invasion of high glucose-induced gestational trophoblast cells via inhibiting PPARγ expression, indicating that CTRP6 may play an important role in GDM [22]. Our previous study demonstrated that CTRP6 expression in the adipose tissue of pregnant rats is significantly correlated with blood glucose levels [23]. Nevertheless, the role and the significance of CTRP6 in GDM have not been clearly elucidated. Therefore, it is necessary to determine the expression profile of CTRP6 during the pathological process of GDM to explore its potential role in GDM. 

In this study, we first measured the plasma levels of CTRP6 in Chinese Han GDM groups and NGT groups at second trimester, third trimester and delivery, and then investigated their relationship with clinical index, metabolic variables and fetal parameters. Next, we examined the expression levels of CTRP6 in the adipose tissue and placenta of GDM and NGT groups from second trimester to parturition. Finally, we evaluated the expression profile of CTRP6 in the plasma, adipose tissue and placenta of GDM mouse models to analyze the similarity of CTRP6 expression in GDM patients and mouse models.

## 2. Materials and Methods

### 2.1. Patients and Samples

Chinese Han women with singleton pregnancies at the First Affiliated Hospital of Xi’an Jiaotong University from January 2021 to February 2022 were invited to participate in this study (informed consent was obtained from all individual participants). The exclusion criteria for this study were individuals with any preexisting type of diabetes, abnormal thyroid function, infection, inflammation or other obstetric complications. The indications for cesarean section were breech position, mid-pelvic stenosis or scarred uterus. According to the recommendations of the International Association of Diabetes and Pregnancy Study Groups, pregnant women were subjected to a 75 g oral glucose tolerance test (OGTT) during the 24th to 28th weeks of pregnancy. GDM was diagnosed in patients with one or more of the abnormal values, including OGTT fasting ≥ 5.1 mmol/L, 1 h ≥ 10.0 mmol/L and 2 h ≥ 8.5 mmol/L.

Fasting plasma samples were collected from three cohorts, including the second trimester (16–24 weeks) group, the third trimester (31–36 weeks) group and the at-delivery group (GDM, *n* = 9; age-paired normal glucose tolerant subjects (NGT), *n* = 10). Abdominal subcutaneous white adipose tissue (sWAT) and omental white adipose tissue (oWAT) were taken from enrolled individuals during cesarean section. Placenta was taken after delivery. These tissues were stored at −80 °C for further study. 

This clinical study was carried out in accordance with the guidelines and approved by the Ethical Committee of First Affiliated Hospital of Xi’an jiaotong University. (Ethics References No: XJTU1AF2020LSK-275). Informed consent was obtained from all the participants.

### 2.2. Animals

Five-week-old female C57BL/6J mice purchased from the Medical Laboratory Animal Center of Xi’an Jiaotong University (Xi’an, China). Mice were housed in an SPF room at 20 ± 2 °C with a 12 h:12 h light-dark cycle. Animals had free access to water and food. After one week of adaptation, mice were randomly divided into two groups according to their diet: standard chow (CHOW) diet (Beijing Keao Xieli Feed, Beijing, China; D12450B, 3.85 kcal/g; 10%, 70% and 20% of calories from fat, carbohydrate and protein, respectively; *n* = 8) or high fat diet (HFD) (Beijing Keao Xieli Feed, Beijing, China; D12451, 4.73 kcal/g; 45%, 35% and 20% of calories from fat, carbohydrate and protein, respectively; *n* = 14). In order to obtain the GDM animal model, HFD was fed 6 weeks prior to mating and throughout pregnancy. Before mating, random blood glucose levels were tested to ensure there was no difference between the two groups. Two female and one male mouse were mated overnight in each cage. The presence of a vaginal plug in the following morning indicated the first 0.5 day of pregnancy (Gestational Day, GD 0.5). Body weight and food intake were measured every day. Energy intake (kcal/week) was calculated as weekly food intake per mouse according to the energy density of the diet (CHOW (3.85 kcal/g), HFD (4.73 kcal/g)). The fasting blood (mice were fasted for 6 h) was collected via a small tail nick at GD 6.5 (the first trimester), GD 12.5 (the second trimester) and GD 17.5 (the third trimester). The interscapular brown adipose tissue (iBAT), inguinal sWAT, oWAT and placental tissue of the pregnant mice were removed during cesarean section at GD 18.5 (*n* = 6 per group), weighed, and then stored at −80 °C for further study. 

All animal experiments were approved by the Institutional Animal Care and Use Committee (IACUC) of Xi’an Jiaotong University (Number: XJTU-2017-01-01-778). We complied with the ARRIVE guidelines.

### 2.3. Glucose Tolerance Test for Pregnant Mice

At GD 14.5, pregnant mice were fasted 6 h and injected intraperitoneally with a saline glucose solution at 1 g/kg body weight. Blood was collected from tail vein and glucose levels were detected by One Touch Sure Step Test Strips (Johnson & Johnson, New Brunswick, NJ, USA) before and 15, 30, 60, 90 and 120 min after injection. 

### 2.4. Measurements of Plasma Lipid

Maternal plasma total triglycerides (TG); total cholesterol, (CHO); high-density lipoprotein cholesterol (HDL-C) and low-density lipoprotein cholesterol (LDL-C) levels were measured using a LABOSPECT 008AS platform (Hitachi High-Tech Co., Tokyo, Japan).

### 2.5. Measurements of CTRP6, Adiponectin and Insulin Levels in Plasma

Plasma levels of CTRP6, adiponectin and insulin were determined by ELISA using commercial ELISA kits (human CTRP6 (No. H407-1); mouse CTRP6 (No. M407-1); mouse adiponectin (No. H179) and mouse insulin (No. H203-1-2) from Nanjing Jiancheng Bioengineering Institute, Nanjing, China) following manufacturer’s protocols. The homeostasis model assessment index for insulin resistance (HOMA-IR) was determined using the following formula: HOMA-IR = (fasting glucose (mmol/L) × fasting insulin (µU/L))/22.5. 

### 2.6. RNA Extraction, cDNA Synthesis, and Quantitative PCR

The total RNA was isolated using Trizol (Invitrogen, San Diego, CA, USA). RT-PCR kits (Thermo scientific, Waltham, MA, USA) were used for inversing transcript cDNA. Quantitative PCR was analyzed using specific primers and SYBR Green I reagent (Takara, Dalian, China). Gene expression changes were calculated with the 2^−ΔΔCt^ method relative to *Gapdh* and *Cyclophilin* housekeeping genes. The results were obtained by real-time qPCR instrument (Bio-Rad). All primers were ordered from AUGCT (Beijing, China); their sequences are shown in Table 1.

### 2.7. Western Blotting Analysis

Previously described procedures were used [24]. Briefly, tissues were homogenized by RIPA lysis buffer (Life technologies, Carlsbad, CA, USA). Then, 30 μg of protein per lane was separated by 10% SDS-PAGE and transferred on polyvinylidene difluoride (PVDF) membrane (Millipore, Bedford, MA, USA). The membranes were blocked with 5% skim milk in 0.1% TBST buffer at room temperature for 1 h, then incubated with primary antibodies (anti-CTRP6 (1:1000) (No. 36900, abcam, Cambridge, UK)) overnight at 4 °C. After incubation with appropriate secondary horseradish peroxidase (HRP)-coupled antibody, the signals were visualized using an Immobilon HRP substrate (Millipore) and detected using ChemiDoc Touch Imaging System (Bio-Rad, Hercules, CA, USA). Densitometry analysis was performed with Image Lab™ Software Version 5.2 (Bio-Rad, Hercules, CA, USA). Normalization was carried out with reference to the total lane protein. 

### 2.8. Statistical Analysis

Data are shown as the mean ± standard deviation, and n represents the number of human or mice samples. Pair-wise differences were tested by Student’s *t*-test. One-way ANOVA with a Tukey’s *post hoc* test was used for multiple comparisons. Correlation of plasma CTRP6 levels with anthropometric and additional characteristics were measured using Pearson analysis. All data were analyzed by statistical software SPSS 19.0 (SPSS, Chicago, IL, USA) and all figures were made using GraphPad Prism 8.0 (GraphPad Software Inc., San Diego, CA, USA). 

## 3. Results

### 3.1. Clinical Characteristics of the Research Cohort

The clinical characteristics of the participants and their newborns are illustrated in Table 2. During the second trimester, women with GDM had a higher body mass index (BMI) and systolic pressure compared with the NGT group. In the third trimester and at-delivery cohort, there were no obvious differences with regard to the maternal age, BMI, levels of hemoglobin (HGB) and white blood cells (WBC) or systolic pressure and diastolic pressure between the GDM and NGT groups. Notably, significant differences were observed in the placenta weight between the GDM and NGT groups.

We further measured the plasma lipids and insulin levels of pregnant women. As shown in Figure 1, there was no significant difference in the expression of triglyceride (TG), high-density lipoprotein cholesterin (HDL-C) and low-density lipoprotein cholesterin (LDL-C) during pregnancy between the GDM group and the NGT group (Figure 1). Perhaps unexpectedly, the data showed that mothers with GDM have lower plasma cholesterol levels during the second trimester (*p*_second trimester_ = 0.0093) (Figure 1). In addition, higher plasma insulin levels were observed in GDM group during the second trimester (*p*_second trimester_ = 0.0324) and the third trimester (*p*_third trimester_ = 0.0460), while no significant difference was observed at delivery (Figure 1). Furthermore, the homeostasis model assessment of insulin resistance (HOMA-IR) revealed that insulin resistance was also higher during the second and third trimesters in the GDM group (*p*_second trimester_ = 0.0303, *p*_third trimester_ = 0.0260) (Figure 1).

### 3.2. Expression Profile of CTRP6 in Plasma, Placenta and Adipose Tissue

We then investigated the plasma levels of CTRP6 during pregnancy. Compared with the NGT group, the maternal plasma CTRP6 levels in the GDM group were significantly lower during the second trimester (*p*_second trimester_ = 0.0268) and third trimester (*p*_third trimester_ = 0.0010), but significantly higher at delivery (*p*_delivery_ = 0.0251) (Figure 2). Interestingly, when we examined the mRNA expression of CTRP6 in the placenta, the higher expression level was observed in the GDM group (*p* < 0.0001) (Figure 2). Meanwhile, the mRNA expression of CTRP6 was also significantly increased in the oWAT and sWAT of mothers with GDM at delivery (*p*_sWAT_ < 0.0001; *p*_oWAT_ = 0.0003) (Figure 2). 

### 3.3. Correlations between Circulating Levels of CTRP6 and Clinical Features

Then, we analyzed the correlation of circulating CTRP6 levels with the clinical characteristics of pregnant women and newborns. As shown in Table 3, in the second trimester, plasma CTRP6 levels were only significantly negatively correlated with diastolic pressure (r = −0.50, *p =* 0.03). Interestingly, in the third trimester, plasma CTRP6 levels were observed to be positively correlated with white blood cell count in peripheral blood (r = 0.48, *p =* 0.04), while negatively correlated with fasting blood glucose (r = −0.52, *p =* 0.02) and one-hour postprandial blood glucose (r = −0.58, *p =* 0.01). 

### 3.4. Changes of CTRP6 Expression in GDM Mice

In order to further explore the role of CTRP6 in GDM, we established the GDM mice model. As expected, GDM mice had higher levels of blood glucose compared to NGT mice at GD 12.5 and GD 17.5 (*p*_GD 12.5_ = 0.0372, *p*_GD 17.5_ = 0.0336) (Figure 3). Interestingly, from GD 6.5 to GD 17.5, plasma insulin levels were higher in the GDM group (*p*_GD 6.5_ = 0.0163, *p*_GD 12.5_ = 0.0153, *p*_GD 17.5_ = 0.0437) (Figure 3). However, no significant difference in the HOMA-IR value was observed between the GDM and NGT groups (*p >* 0.05) (Figure 3). Intraperitoneal GTT was performed at GD 14.5; the GDM group exhibited significantly higher glucose levels at 15 min and 30 min after the injection of glucose (*p*_GD 6.5_ = 0.0107, *p*_GD 12.5_ = 0.0033) and a larger blood glucose area under the curve (AUC) compared with the NGT mice (*p =* 0.0105) (Figure 3). In addition, plasma CTRP6 levels in GDM mice at GD 6.5 and GD 12.5 were lower than in NGT mice (*p*_GD 6.5_ = 0.0274, *p*_GD 12.5_ = 0.0077), but no significant difference was observed at GD 17.5 (*p >* 0.05) (Figure 3). Interestingly, levels of adiponectin in the GDM group were lower than in the NGT group (*p*_GD 17.5_ = 0.0165) (Figure 3).

GDM mice underwent cesarean section at GD 18.5. As expected, the gestational weight gain in the GDM mice was higher than that in the NGT group (*p =* 0.0173) (Figure 4). Correspondingly, increased sWAT and visceral WAT (vWAT) mass and body fat ratio was observed in GDM mice (*p*_sWAT_ = 0.0044, *p*_vWAT_ = 0.0068, *p_body fat_*
_ratio_ = 0.0202) (Figure 4). However, regarding the result of neonatal number and weight, there was no significant difference between the GDM and NGT groups (*p >* 0.05) (Figure 4). 

Next, we investigated the changes in CTRP6 expression in GDM mice. Consistent with the expression changes in GDM patients, plasma levels of CTRP6 were increased in GDM mice at delivery (*p =* 0.0019) (Figure 5). The mRNA levels of *Ctrp6* in the sWAT (_sWAT_ = 0.0153) and vWAT (*p*_vWAT_ = 0.0279) of GDM mice were significantly higher, but they were lower in the iBAT (*p*_iBAT_ = 0.0255) and not significantly different in the placenta (Figure 5). Moreover, CTRP6 protein expression in the iBAT (*p*_iBAT_ = 0.0476) was significantly lower in the GDM group (Figure 5).

## 4. Discussion

Current research has revealed that adipokine dysregulation is pivotal in the pathogenesis of GDM. Abnormal expression of adipokines including adiponectin, leptin and visfatin might have pathological significance and a prognostic value in this pregnancy complication [9]. In the present study, we examined the plasma levels and tissue-specific expressions of CTRP6 during pregnancy. Our study presents the first evidence that circulating levels of CTRP6 dynamically change during pregnancy in GDM patients relative to healthy individuals. CTRP6 levels showed a significant association with the parameters involved in GDM pathogenesis.

Previous studies suggested that CTRP6, a member of adiponectin gene family, is involved in diverse physiological and pathophysiological processes by regulating cell proliferation and apoptosis, cell differentiation, insulin sensitivity and chronic inflammatory response [15,20,25,26]. Some clinical studies have revealed that serum CTRP6 levels are significantly elevated in overweight, obese and T2DM individuals. Most importantly, receiver operating characteristic (ROC) curve analysis showed that CTRP6 levels were associated with insulin resistance [19,27]. In addition, a previous study found that CTRP6 knockdown improved insulin sensitivity and reduced diet-induced obesity in mice [21]. Mechanistically, excessive CTRP6 aggravates insulin resistance by downregulating the phosphorylation of Akt and p38MAPK, while upregulating the Hedgehog signaling pathway in adipose tissue [20,21]. Although during normal pregnancy, in order to provide maximum nutrition for fetal development, the mother exhibits physiological insulin resistance, pathological insulin resistance is the typical characteristic of GDM patients [28,29]. However, it has not been reported whether the expression level of CTRP6 is an abnormal expression in GDM patients at different stages of pregnancy. Our finding of the increased circulating CTRP6 levels at delivery in GDM patients and mice models is in line with a previous study that found significantly increased CTRP6 levels in T2DM individuals [19]. However, we also observed that circulating CTRP6 levels were decreased during the second and third trimesters in GDM patients and during the first and second trimesters in GDM mice. Indeed, GDM is usually diagnosed in the second trimester, when pregnant women already show glucose intolerance and elevated plasma insulin levels as indicated by HOMA-IR assessment. However, as the type of diabetes first diagnosed during pregnancy, GDM has many differences compared to T2DM. For example, the condition is mediated by placenta factors in GDM patients, and the development process of GDM is shorter than that of T2DM (most GDM patients can recover within weeks after delivery) [30]. In this study, we observed that although GDM patients still show glucose intolerance at delivery, plasma insulin and HOMA-IR have returned to normal levels. Therefore, the decreased CTRP6 in first or the second and third trimesters is speculated to be due to the negative-feedback protection mechanism of initial blood glucose intolerance in GDM individuals. This inference is further supported by the negative correlation between plasma CTRP6 levels and OGTT blood glucose levels at 0 and 1 h. This suggests that the decrease of circulating CTRP6 in early pregnancy can serve as an early biomarker for the diagnosis of GDM. However, further studies are necessary to determine the underlying mechanisms. 

Adipose tissue remodeling is an adaptive response to pregnancy that includes the anabolic phase in first trimester, fat-storing state in the second trimester and catabolic state in the third trimester [31]. However, excessive accumulation of adipose tissue leads to overweight or obesity, which is one of the important risk factors for GDM [30]. Consistent with this, our study showed that the BMI of GDM patients in the second trimester was significantly higher than that of the NGT group. Interestingly, we also saw a significant increase in gestational weight gain and body fat ratio in GDM mouse models. As an adipokine, the main source of plasma CTRP6 is adipose tissue. Our studies support this view as we observed increased CTRP6 at the mRNA level in the sWAT and oWAT of GDM patients, which is consistent with the high expression of plasma CTRP6 in GDM patients at delivery. It is reported that CTRP6 promotes the differentiation of 3T3-L1 preadipocytes by promoting key genes in adipogenesis and the Erk1/2 signaling pathway [32]. Another study in porcine preadipocytes found that CTRP6 promotes adipocyte differentiation through the MAPK signaling pathway [33]. Our data show that, compared to normal pregnant women, GDM patients have a higher BMI, accompanied by an abnormal increase in CTRP6 expression. Does this suggest that the high expression of CTRP6 in the adipose tissue of GDM patients accelerates fat formation, thereby increasing BMI? Such a hypothesis still needs to be further studied. Meanwhile, we also observed an increase in *Ctrp6* gene expression in the sWAT and vWAT of GDM mice, which was consistent with the above result, but an increase in CTRP6 protein was not observed. This discrepancy might be attributed to the pre-translational control in mRNA [34] or the stability of the protein before secretion. However, the specific mechanisms still require further study. Interestingly, mRNA and protein expression of CTRP6 exhibited a significant decrease in the iBAT of GDM mice. Given that *Ctrp6* deficiency has been shown to increase the expression of thermogenic markers in brown adipocytes [21], the reduction in CTRP6 in the iBAT of GDM mice could be the compensatory mechanism. Further studies on the function of CTRP6 in thermogenesis are necessary.

Emerging evidence suggests that CTRP6 is also highly expressed in placenta [13,17,22]. Furthermore, Zhang Jin et al. reported that CTRP6 was up-regulated in high glucose-induced trophoblast cells [22]. Consistent with this, our study showed that with the increase in placental weight in GDM patients, the expression of CTRP6 was up-regulated. However, we did not observe any changes in the expression level of *Ctrp6* in the placenta of GDM mice. This result suggests that it is best to choose the human sample to study the role of CTRP6 in GDM placenta. 

Nevertheless, our study possesses a few limitations. Firstly, because this study required the collection of blood samples during the whole pregnancy, and a part of adipose tissue during caesarean section, the number of participants in this study is small. In order to increase the sample size and further study the regulatory role of CTRP6 in GDM patients, we will continue to apply for funding and timely ethical approval in the future. Secondly, due to the provisions of animal ethics documents, we were unable to collect adipose tissue and placenta from GDM mice during the first, second and third trimesters of pregnancy, which limited our exploration of the dynamic alteration of CTRP6 in the main tissues throughout pregnancy. 

In summary, we demonstrated the dynamic expression profile of circulating CTRP6 during pregnancy. The expression profile of CTRP6 during pregnancy is consistent between GDM human and mice. CTRP6 was over-expressed in plasma, adipose and placenta in GDM humans and mice at delivery. Collectively, these observations suggest that CTRP6 may serve as a candidate biomarker for the treatment of GDM. 

## Figures and Tables

**Figure 1 biomedicines-12-01128-f001:**
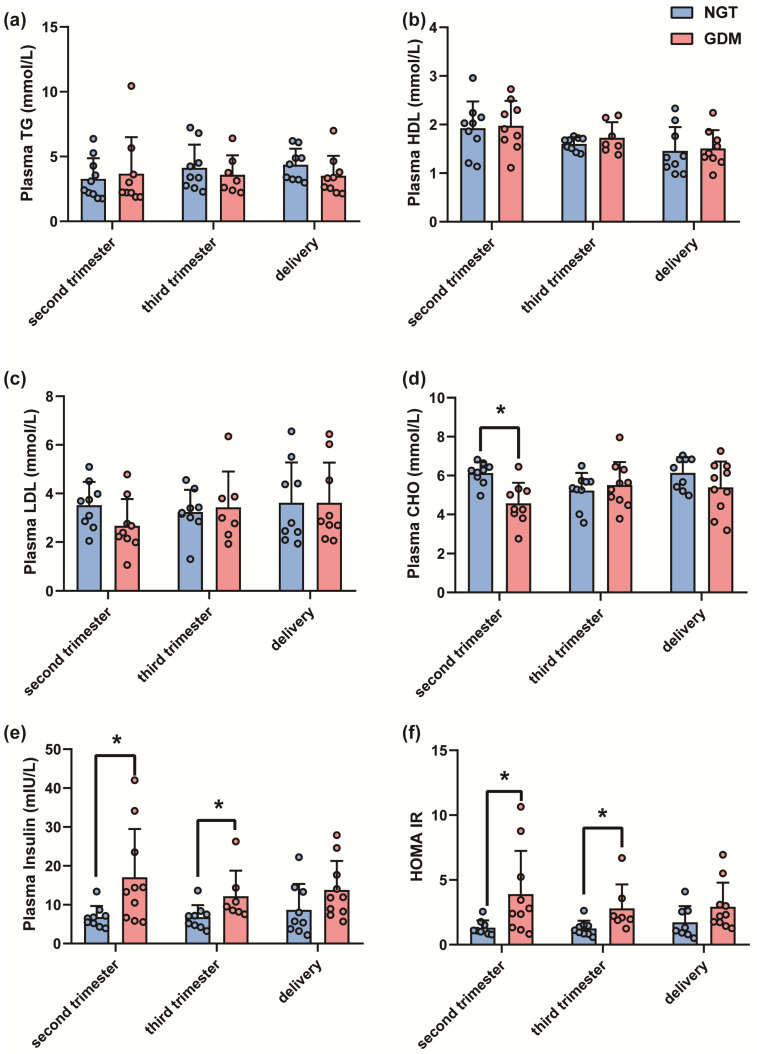
The plasma lipid and glycemic outcomes were significantly changed in pregnant women with GDM compared to NGT. (**a**) Plasma level of TG. (**b**) Plasma level of CHO. (**c**) Plasma level of LDL-C. (**d**) Plasma level of HDL-C. (**e**) Plasma level of insulin. (**f**) HOMA-IR index (GDM, *n* = 9; NGT, *n* = 10). All data are shown as mean ± SD, * *p* < 0.05 vs. NGT.

**Figure 2 biomedicines-12-01128-f002:**
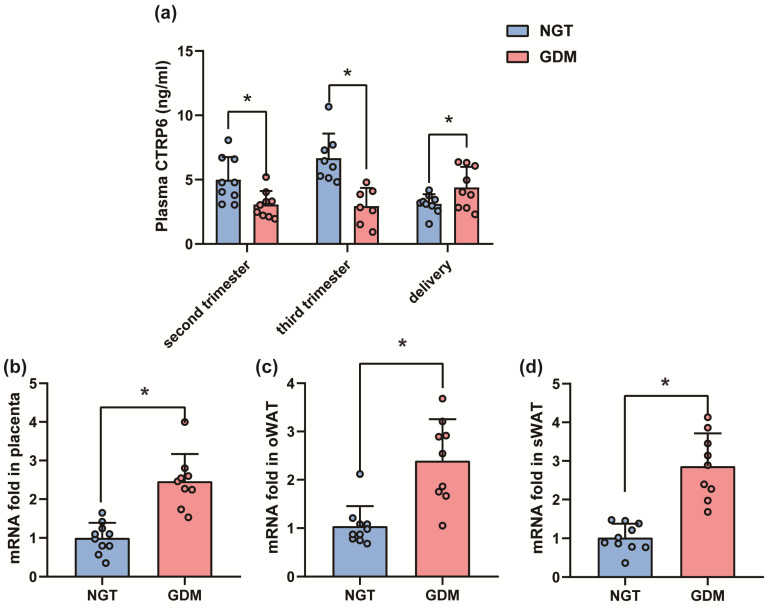
CTRP6 expression levels were altered in pregnant women with GDM compared to NGT. (**a**) Plasma CTRP6 levels (GDM, *n* = 9; NGT, *n* = 10). (**b**–**d**) Ctrp6 mRNA fold in placenta, oWAT, sWAT of pregnant women at caesarean delivery (GDM, *n* = 9; NGT, *n* = 10). All data are shown as mean ± SD, * *p* < 0.05 vs. NGT.

**Figure 3 biomedicines-12-01128-f003:**
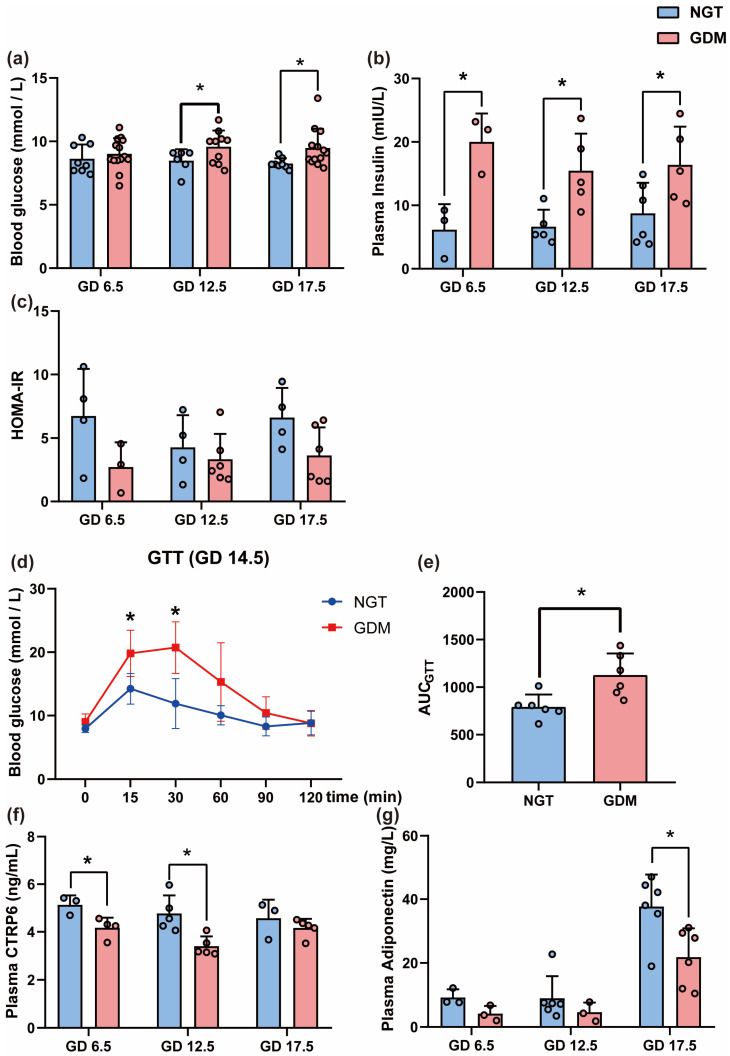
The glycemic outcomes, plasma CTRP6 and adiponectin levels of GDM mouse models at different pregnancy stages. (**a**) Fasting blood glucose (GDM, *n* = 8; NGT, *n* = 14). (**b**) Plasma insulin level. (**c**) HOMA-IR index. (**d**) GTT. (**e**) GTT AUC. (**f**) Plasma CTRP6 level. (**g**) Plasma adiponectin level. *n* = 6 per group. All data are shown as mean ± SD, * *p* < 0.05 vs. NGT.

**Figure 4 biomedicines-12-01128-f004:**
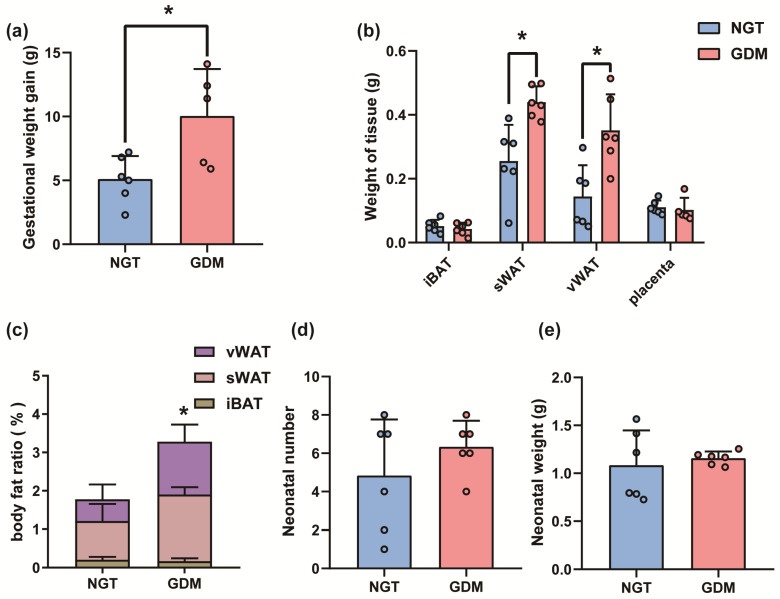
The basic characteristics of GDM mouse models at caesarean delivery. (**a**) Gestational weight gain. (**b**) Weight of iBAT, sWAT, vWAT and placenta. (**c**) Body fat ratio. (**d**) neonatal number. (**e**) neonatal weight. *n* = 6 per group. All data are shown as mean ± SD, * *p* < 0.05 vs. NGT.

**Figure 5 biomedicines-12-01128-f005:**
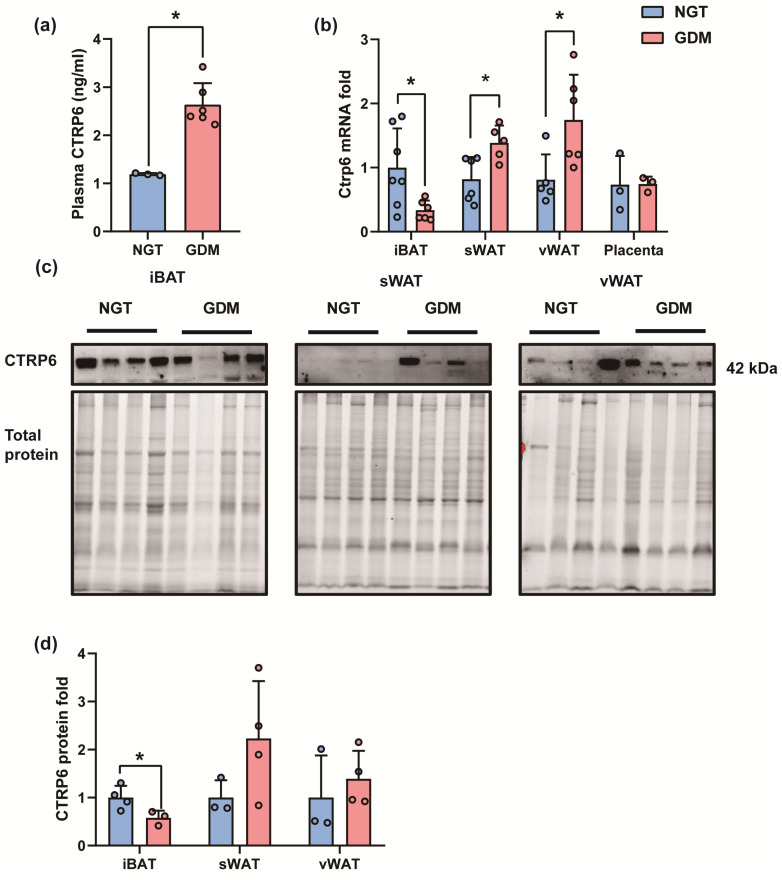
CTRP6 expression profile in GDM mouse models at caesarean delivery. (**a**) Plasma level of CTRP6. (**b**) Ctrp6 mRNA fold in iBAT, sWAT, vWAT and placenta. (**c**) Western blot analysis of CTRP6. (**d**) CTRP6 protein fold in iBAT, sWAT and vWAT. All data are shown as mean ± SD, * *p* < 0.05 vs. NGT.

**Table 1 biomedicines-12-01128-t001:** The primer sequences for Real-time quantitative PCR analysis.

Gene	Species	Accession Number	Primer Sequences (from 5′ to 3′)	Production Size
*Ctrp6*	*Mus musculus*	NM_028331	Forward primer: CCTTATGTCCTGCCTGAAGTCAG	144
			Reverse primer: ACCTTTGACACCCTGAGAGCCA	
*Gapdh*	*Mus musculus*	NM_008084	Forward primer: ACTGAGGACCAGGTTGTC	136
			Reverse primer: TGCTGTAGCCGTATTCATTG	
*Cyclophilin*	*Mus musculus*	NM_008907	Forward primer: CATACAGGTCCTGGCATCTTGTC	112
			Reverse primer: AGACCACATGCTTGCCATCCAG	
*CTRP6*	*Homo sapiens*	NM_031910	Forward primer: CTGCCTGAGATCAGACCCTACA	140
			Reverse primer: TTGTCACCCTTGCTGCCCTGAG	
*GAPDH*	*Homo sapiens*	NM_002046	Forward primer: GTCTCCTCTGACTTCAACAGCG	131
			Reverse primer: ACCACCCTGTTGCTGTAGCCAA	
*Cyclophilin*	*Homo sapiens*	NM_021130	Forward primer: GGCAAATGCTGGACCCAACACA	161
			Reverse primer: TGCTGGTCTTGCCATTCCTGGA	

**Table 2 biomedicines-12-01128-t002:** Baseline characteristics of participants.

Characteristic	Second Trimester		Third Trimester		Delivery	
	NGT	GDM	NGT	GDM	NGT	GDM
Maternal Parameters						
Age (year)	31.50 ± 3.66	31.56 ± 3.97	31.70 ± 1.16	32.86 ± 4.95	29.89 ± 3.98	30.67 ± 3.84
BMI (kg/m^2^)	21.56 ± 2.75	24.95 ± 3.24 *	23.94 ± 2.23	25.29 ± 2.81	27.86 ± 3.03	29.75 ± 3.56
HGB (g/L)	117.10 ± 9.30	122.56 + 9.76	120.00 ± 14.13	121.22 ± 13.18	115.20 ± 9.48	115.22 ± 19.31
WBC (10^9^/L)	8.36 ± 1.84	10.34 ± 2.53	10.57 ± 3.37	8.92 ± 2.56	9.28 ± 2.20	12.33 ± 6.05
Systolic pressure (mmHg)	102.70 ± 7.89	113.11 ± 9.49 *	107.40 ± 8.00	113.00 ± 9.37	115.70 ± 13.87	122.44 ± 12.28
Diastolic pressure (mmHg)	69.90 ± 5.93	72.56 ± 4.98	69.50 ± 6.85	75.89 ± 3.10	81.20 ± 10.56	79.44 ± 8.59
OGTT 0 h (mmol/L)	4.23 ± 0.37	5.03 ± 0.52 *	4.06 ± 0.34	5.09 ± 0.91 *	4.16 ± 0.37	4.74 ± 0.68 *
OGTT 1 h (mmol/L)	6.31 ± 1.24	10.31 ± 0.76 *	6.59 ± 0.88	10.99 ± 1.21 *	6.72 ± 0.83	10.61 ± 1.59 *
OGTT 2 h (mmol/L)	5.69 ± 0.84	8.02 ± 1.67 *	5.69 ± 0.82	8.49 ± 1.29 *	5.79 ± 0.63	8.46 ± 1.22 *
Neonatal parameters						
Birth height (cm)					50.00 ± 0.58	48.40 ± 3.13
Birth weight (g)	/	/	/	/	3338.00 ± 412.40	3557.56 ± 488.73
Blood glucose (mmol/L)	/	/	/	/	3.48 ± 0.61	3.50 ± 0.67
Placenta (g)	/	/	/	/	511.00 ± 35.42	575.56 ± 46.40 *

Data are shown as mean ± standard deviation, NGT: normal glucose tolerance, GDM gestational diabetes mellitus, BMI: body mass index, HGB: hemoglobin, WBC: white blood cell, OGTT: oral glucose tolerance test, *: *p* < 0.05.

**Table 3 biomedicines-12-01128-t003:** Pearson correlation analysis investigating the association between plasma CTRP6 levels and clinical indicators.

Correlation	Second Trimester		Third Trimester		Delivery	
	r	*p*	r	*p*	r	*p*
Maternal indices						
Age	−0.27	0.27	−0.06	0.82	0.42	0.08
BMI	−0.08	0.76	−0.24	0.32	−0.20	0.42
HGB	−0.08	0.75	0.02	0.41	0.11	0.66
WBC	−0.11	0.66	0.48	0.04 *	0.19	0.44
Systolic pressure	−0.50	0.03 *	−0.32	0.18	−0.20	0.41
Diastolic pressure	−0.01	0.96	−0.41	0.08	−0.19	0.44
OGTT 0 h	−0.34	0.16	−0.52	0.02 *	0.17	0.50
OGTT 1 h	−0.25	0.29	−0.58	0.01 *	0.35	0.14
OGTT 2 h	−0.27	0.26	−0.40	0.09	0.26	0.29
CHO	−0.10	0.70	−0.23	0.34	−0.16	0.53
Neonatal parameters						
Birth height (cm)					−0.40	0.23
Birth weight (g)	/	/	/	/	0.42	0.41
Blood glucose (mmol/L)	/	/	/	/	−0.30	0.21
Placenta (g)	/	/	/	/	0.42	0.08

BMI: body mass index, HGB: hemoglobin, WBC: white blood cell, OGTT: oral glucose tolerance test, CHO: cholesterol, *: *p* < 0.05.

## Data Availability

Data are contained within the article.

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
