# Peer review of "Characterizing the Dynamic Expression of C1q/TNF-α-Related Protein 6 (CTRP6) during Pregnancy in Humans and Mice with Gestational Diabetes Mellitus"

_biomedicines, 2024, doi:10.3390/biomedicines12051128_

Round 1
Reviewer 1 Report
Comments and Suggestions for Authors
Dear Authors,
my comments:
1. In my opinion title of the article is not adequate to content.
2. Why did you decide to perform research on human and mice?
3. In my opinion English revision is needed
Author Response
We appreciate your constructive comments concerning our manuscript. The specific responses are listed below:
- In my opinion title of the article is not adequate to content.
We appreciate your observation about the title. We agree that the original title may not fully encapsulate the scope and findings of our research. We have revised the title to better reflect the content and significance of our study. The new title is: Characterizing the Dynamic Expression of C1q/TNF-α-related protein 6 (CTRP6) during Pregnancy in Human and Mouse with Gestational Diabetes Mellitus. You can also find it on Title page, highlighted in red.
- Why did you decide to perform research on human and mice?
We selected both human and mice models for our research to leverage the strengths of each system. Human data provide direct relevance to clinical conditions, while mice models allow us to manipulate genetic and environmental factors to understand underlying mechanisms. This comparative approach strengthens our findings by allowing us to validate mechanisms observed in human subjects through controlled experimental conditions in mice.
- In my opinion English revision is needed.
We acknowledge the importance of clear and precise language, especially in scientific communication. We have engaged a professional language editing service to thoroughly review and refine the manuscript. This additional layer of revision has improved the overall readability and has ensured that our research is presented in the most articulate manner.
Reviewer 2 Report
Comments and Suggestions for Authors
Comments:
Introduction:
The introduction laid a great background for the study by outlining the importance of adipokine dysregulation in GDM; however, it may be helpful to have a more in-depth literature review regarding CTRP6 in general pregnancy and specifically GDM. It should clearly state the research gap with strong rationale of why there is a need for this study and how further this work will help advance in understanding GDM pathophysiology.
Methodology:
The description of patient selection, sample collection, and experimental design for human and mouse models is to be done. Make sure that the paper describes clearly, for both human and animal subjects of the study, the description of inclusion and exclusion criteria and ethical considerations. The methods should describe the controls applied, the statistical models used, and any methodological limitation, such as whether the timing of the sample collection corresponds to dynamic changes in the levels of these adipokines over the course of pregnancy.
Results:
· The obtained results are very exciting with the dynamic expression pattern of CTRP6. Whether the changes of CTRP6 levels are the cause or the effect of metabolic changes in GDM, this has to be related.
· A negative correlation between the CTRP6 level and glucose tolerance test would, therefore, suggest a functional role of CTRP6 in glucose homeostasis. However, to mention here, the possibility of confounding and that longitudinal studies would provide clarity in that direction. All of these differences together, because of this, would suggest that the observed differences between the mRNA and protein levels of CTRP6, indeed, are most likely due to post-transcriptional modifications and stability of the protein before being secreted.
Discussion:
· The authors well discussed the results obtained with available literature in regard to adipokine dysfunction in metabolic diseases; however, there needs to be further discussion about the mechanistic role of CTRP6 in insulin resistance and a crystal clear view as a potential therapeutic target. What are the implications of GDM patients having higher levels of CTRP6 at delivery? Is this response protective, or is it mirroring pathological changes? Interestingly, CTRP6 expression was found upregulated in both sWAT and oWAT of the GDM women. Its patho-physiological implications with respect to this tissue specificity and relevance to the insulin resistant state of GDM need to be elaborated.
· The reduction of CTRP6 in iBAT expression should receive importance in view of its possible compensatory mechanisms that may be available in GDM, as brown adipose tissue is involved in energy expenditure.
Conclusions:
This underscores the potential of CTRP6 as a biomarker for GDM. However, one does expect the conclusion of this article to carry a word of caution on the necessity of furthering research to appreciate the causative relationship between CTRP6 levels and GDM.
Overall Assessment:
The present research brings some valuable insights but also raises several unanswered questions. Derived conclusions were based on such findings and, therefore, were taken tentatively to emphasize their preliminary character. Multi-center longitudinal sampling of large numbers, and functional experiments that may be able to elucidate the biological role of CTRP6 in GDM, could form a research direction. Finally, questions should be directed at how such findings may be applied in clinical practice and what further steps are needed to move from correlation to a clearer understanding of causation.
Author Response
Responses to the reviewers’ comments:
We appreciate your constructive comments concerning our manuscript. The specific responses are listed below:
Reviewer 2
Comments:
Introduction:
The introduction laid a great background for the study by outlining the importance of adipokine dysregulation in GDM; however, it may be helpful to have a more in-depth literature review regarding CTRP6 in general pregnancy and specifically GDM. It should clearly state the research gap with strong rationale of why there is a need for this study and how further this work will help advance in understanding GDM pathophysiology.
Thank you for your insightful comments and suggestions for enhancing the introduction of our manuscript. We have taken your advice and conducted a more in-depth literature review to address the research gap regarding CTRP6 in general pregnancy and specifically in the context of GDM.
Upon further research, we identified a relevant study about placental trophoblast cells in GDM, which discusses the role of CTRP6 in regulating the vitality, migration, and invasiveness of trophoblast cells under hyperglycemic conditions via the PPARγ signaling pathway (PMID: 34964705). We have included this reference (ref.22) in our introduction to provide a more comprehensive background on CTRP6's relevance to GDM.
Additionally, to elucidate the broader functions of CTRP6, we have incorporated recent findings regarding its role in the alternative complement pathway, extracellular matrix, and recurrent miscarriage. These additional references have been integrated into the manuscript at Line 69-76, highlighted in red for easy reference.
We believe that these revisions not only strengthen the rationale for our study but also contribute to a deeper understanding of the pathophysiology of GDM. We are grateful for the opportunity to refine our work and hope that our responses meet with your approval.
Methodology:
The description of patient selection, sample collection, and experimental design for human and mouse models is to be done. Make sure that the paper describes clearly, for both human and animal subjects of the study, the description of inclusion and exclusion criteria and ethical considerations. The methods should describe the controls applied, the statistical models used, and any methodological limitation, such as whether the timing of the sample collection corresponds to dynamic changes in the levels of these adipokines over the course of pregnancy.
Thank you for your detailed comments and for highlighting the need for clarity in our methodology section. We have made the following revisions to ensure that our methods are transparent and comprehensive:
We have expanded our description to include a detailed account of the criteria used for mouse models (Line 132-134 of revision manuscript, highlighted in red), ensuring that the methodology is replicable and the selection process is clear to readers.
To better present the ethical considerations, we have displayed the ethical approval number for the clinical experiments (Line 118-121) and for the animal experiments (Line 145-147) under their respective sections. Please find the revised manuscript where these numbers are highlighted in red at the corresponding locations.
Regarding the timing of sample collection, in the mouse experiments, samples were uniformly collected at a specified time to ensure consistency. As for our clinical trial participants, samples and clinical information were strictly collected according to the designated timing at each stage of their involvement, starting from the time of enrollment. Each participant's information is accurately matched and corresponded. Due to this rigorous approach, the number of clinical samples we obtained is relatively small. We have also analyzed the reasons for this limitation in the discussion section of this manuscript (Line 357-359).
We believe that these revisions have significantly improved the clarity and rigor of our methodology section. We appreciate your guidance and are confident that our responses are in line with the high standards of scientific reporting.
Results:
- The obtained results are very exciting with the dynamic expression pattern of CTRP6. Whether the changes of CTRP6 levels are the cause or the effect of metabolic changes in GDM, this has to be related.
Thank you for your insightful comments. Indeed, the observation that CTRP6 levels fluctuate significantly within the GDM population raises important questions about the nature of this relationship. Following your reminder, we will proceed with the following experiments in our subsequent research: a longitudinal study to track CTRP6 levels throughout pregnancy with more participants enrolled, correlation analysis with metabolic markers, and an investigation into the potential of CTRP6 as a biomarker for early detection and as a therapeutic target. Thanks again for your advice.
- A negative correlation between the CTRP6 level and glucose tolerance test would, therefore, suggest a functional role of CTRP6 in glucose homeostasis. However, to mention here, the possibility of confounding and that longitudinal studies would provide clarity in that direction. All of these differences together, because of this, would suggest that the observed differences between the mRNA and protein levels of CTRP6, indeed, are most likely due to post-transcriptional modifications and stability of the protein before being secreted.
Thank you for your insightful comments. The differences are most likely due to post-transcriptional modifications and the stability of the protein before secretion is well taken. We agree that these factors could significantly contribute to the observed variance. We have also added this inference in the discussion section of the revised manuscript. Please find Line 345-347 of the revised manuscript, where it is highlighted in red.
Discussion:
- The authors well discussed the results obtained with available literature in regard to adipokine dysfunction in metabolic diseases; however, there needs to be further discussion about the mechanistic role of CTRP6 in insulin resistance and a crystal clear view as a potential therapeutic target. What are the implications of GDM patients having higher levels of CTRP6 at delivery? Is this response protective, or is it mirroring pathological changes? Interestingly, CTRP6 expression was found upregulated in both sWAT and oWAT of the GDM women. Its patho-physiological implications with respect to this tissue specificity and relevance to the insulin resistant state of GDM need to be elaborated.
We appreciate your suggestion for a more in-depth discussion on the mechanistic role of CTRP6 in insulin resistance and its potential as a therapeutic target. To provide a more comprehensive perspective, we have expanded our literature review to include additional studies on the role of CTRP6 in metabolic diseases, with a focus on its involvement in insulin resistance. We have incorporated a discussion of the mechanisms into the discussion section of revised manuscript, which can be found on Line 309-311.
Regarding the higher circulation level of CTRP6 and upregulated expression in sWAT and oWAT of the GDM women at delivery, we speculate that this is a pathological change, and this trend is consistent with the changes observed in patients with obesity or type 2 diabetes. We have also mentioned this viewpoint in our discussion section of revised manuscript, Line 344-351, 355-357, highlighted in red.
- The reduction of CTRP6 in iBAT expression should receive importance in view of its possible compensatory mechanisms that may be available in GDM, as brown adipose tissue is involved in energy expenditure.
Thank you for your insightful comment regarding the potential significance of CTRP6 expression in iBAT in the context of GDM. We have incorporated this analysis into the discussion section as you suggested(Line 356-360). Your suggestion has reminded us that future research could focus on the impaired thermogenic capacity of brown adipose tissue in patients with GDM. We are grateful once again for your valuable suggestion.
Conclusions:
This underscores the potential of CTRP6 as a biomarker for GDM. However, one does expect the conclusion of this article to carry a word of caution on the necessity of furthering research to appreciate the causative relationship between CTRP6 levels and GDM.
Thank you for your thoughtful comments and for highlighting the importance of a cautious interpretation of our findings regarding CTRP6 as a potential biomarker for GDM. We fully agree with your point that while our results are suggestive, they do not establish a causative link between CTRP6 levels and the development or progression of GDM. We will obtain more direct evidence in our future studies.
Overall Assessment:
The present research brings some valuable insights but also raises several unanswered questions. Derived conclusions were based on such findings and, therefore, were taken tentatively to emphasize their preliminary character. Multi-center longitudinal sampling of large numbers, and functional experiments that may be able to elucidate the biological role of CTRP6 in GDM, could form a research direction. Finally, questions should be directed at how such findings may be applied in clinical practice and what further steps are needed to move from correlation to a clearer understanding of causation.
Thank you for your constructive critique of our research. We acknowledge that while our study has provided some valuable insights, it has also raised several questions that remain unanswered. We concur that the conclusions drawn from our findings are tentative, reflecting the preliminary nature of our observations.
Your suggestion for a multi-center, longitudinal study with a large sample size is well-taken. Such a study would indeed enhance the generalizability of our results and provide a more robust platform for understanding the dynamics of CTRP6 in the context of GDM. We also recognize the importance of conducting functional experiments to clarify the biological role of CTRP6 in GDM, which could be a significant direction for future research, like loss-of-function and gain-of-function of CTRP6 in GDM.
Furthermore, we appreciate your emphasis on the translational potential of our findings. We agree that it is crucial to consider how our research might be applied in clinical practice. This includes identifying the necessary steps to move from correlation to causation, which is a complex but essential process in advancing our understanding of GDM.
We are committed to addressing these points in our future work and to contributing to a more comprehensive understanding of the role of CTRP6 in GDM. Your comment has been invaluable in guiding our research direction and ensuring that our conclusions are presented with the appropriate level of caution.
Reviewer 3 Report
Comments and Suggestions for Authors
The article is interesting but not linear. The authors do not clearly describe the main, known and certain functions of the protein, for example on the alternative complement pathway, on the role in the extracellular matrix, which could explain the physiological role in the reproductive system. The somewhat confusing data collected does not exclude that this correlation with the risk factor of obesity and diabetes is secondary to physiological dynamics other than the direct metabolism of glucose and lipogenesis. I suggest reviewing the literature and considering the physiological roles already known present in proteomics databases. Furthermore, the protein is expressed in the endometrium and in the non-villous trophoblast, it could provide a tissue immunohistochemical evaluation and a simpler and more direct evaluation also from cord blood
Author Response
Responses to the reviewers’ comments:
We appreciate your constructive comments concerning our manuscript. The specific responses are listed below:
Reviewer 3
The article is interesting but not linear. The authors do not clearly describe the main, known and certain functions of the protein, for example on the alternative complement pathway, on the role in the extracellular matrix, which could explain the physiological role in the reproductive system.
Thank you for the constructive suggestion. We understand your concern about the need for a more comprehensive description of the CTRP6's known functions, particularly in relation to the alternative complement pathway and its role in the extracellular matrix, as well as its physiological role in the reproductive system. Therefore, we revise the manuscript to include a more detailed discussion. Please find them on Line 69-76.
The somewhat confusing data collected does not exclude that this correlation with the risk factor of obesity and diabetes is secondary to physiological dynamics other than the direct metabolism of glucose and lipogenesis. I suggest reviewing the literature and considering the physiological roles already known present in proteomics databases.
Thank you for your insightful comments regarding the complexity of the data and the potential for secondary physiological dynamics influencing the observed correlation between obesity, diabetes risk factors, and our findings. We agree that it is important to consider a broader range of factors that may impact these relationships. In response to your suggestion, we conduct a thorough review of the existing literature to identify and discuss additional physiological roles that may be relevant to our study. We added the summarized issues to the corresponding sections of the discussion in our revised manuscript (Line 309-311, 344-351, 355-357). We appreciate your guidance in helping us to refine our analysis and interpretation of the data.
Furthermore, the protein is expressed in the endometrium and in the non-villous trophoblast, it could provide a tissue immunohistochemical evaluation and a simpler and more direct evaluation also from cord blood.
Thank you for your kindly suggestion. Regrettably, at the time of the original experimental design, we did not collect cord blood, and placental tissue samples were not fixed for immunohistochemical analysis. Given that the data for this study are derived from one-to-one corresponding samples, we currently do not have the capability to supplement this part of the experiment.
However, your suggestion serves as a valuable reminder for us. In future experiments, we will consider incorporating immunohistochemical detection of CTRP6 expression in placental tissue and using an ELISA method to assess CTRP6 levels in cord blood.
We appreciate your guidance and acknowledge the importance of exploring these avenues to further our understanding of CTRP6's role in the context of GDM.
Round 2
Reviewer 2 Report
Comments and Suggestions for Authors
The authors have sufficiently answered my queries.
Reviewer 3 Report
Comments and Suggestions for Authors
The authors must also report this consideration in the final discussion "Furthermore, the protein is expressed in the endometrium and in the non-villous trophoblast, it could provide a tissue immunohistochemical evaluation and a simpler and more direct evaluation also from cord blood.
Thank you for your kindly suggestion. Regrettably, at the time of the original experimental design, we did not collect cord blood, and placental tissue samples were not fixed for immunohistochemical analysis. Given that the data for this study are derived from one-to-one corresponding samples, we currently do not have the capability to supplement this part of the experiment.
However, your suggestion serves as a valuable reminder for us. In future experiments, we will consider incorporating immunohistochemical detection of CTRP6 expression in placental tissue and using an ELISA method to assess CTRP6 levels in cord blood.
We appreciate your guidance and acknowledge the importance of exploring these avenues to further our understanding of CTRP6's role in the context of GDM."